# Model-Guided Manufacturing of Transducer Arrays Based on Single-Fibre Piezocomposites

**Martin Angerer *** **, Michael Zapf, Benjamin Leyrer and Nicole V. Ruiter**

Karlsruhe Institute of Technology, Institute for Data Processing and Electronics,
76344 Eggenstein-Leopoldshafen, Germany; Michael.Zapf@kit.edu (M.Z.); Benjamin.Leyrer@kit.edu (B.L.);
Nicole.Ruiter@kit.edu (N.V.R.)
* Correspondence: Martin.Angerer@kit.edu; Tel.: +49-721-608-24114

**Abstract:** For breast cancer imaging, ultrasound computer tomography (USCT) is an emerging technology. To improve the image quality of our full 3-D system, a new transducer array system (TAS) design was previously proposed. This work presents a manufacturing approach which realises this new design. To monitor the transducer quality during production, the electro-mechanical impedance (EMI) was measured initially and after each assembly step. To evaluate the measured responses, an extended Krimholtz–Leedom–Matthaei (KLM) transducer model was used. The model aids in interpreting the measured responses and presents a useful tool for evaluating parasitic electric effects and attenuation at resonance. For quality control, the phase angle at thickness resonance $\varphi_t$ was found to be the most specific EMI property. It can be used to verify the functionality of the piezocomposites and allows reliable detection of faults in the acoustic backing. Evaluating the final response of 68 transducers showed 5% variance of the series resonance frequency. This indicates good consistency of derived ultrasound performance parameters.

**Keywords:** transducer array manufacturing; single-fibre piezocomposites; KLM model; electro-mechanical impedance; quality control; ultrasound computer tomography

## 1. Introduction

The current gold standard for early breast cancer imaging is mammography. However, the exposure of radiation and a limited effectiveness for dense breasts are among several disadvantages of this procedure [1,2]. USCT can be a suitable imaging alternative. There, different tissue types can be distinguished due to differences in speed of sound and attenuation [3,4]. Conventional ultrasound devices such as hand-held probes produce anisotropic point spread functions and often lack sufficient resolution. This led to first tomographic approaches in which images are reconstructed based on 3D data acquisition [5].

At the Karlsruhe Institute of Technology, our group proposed a 3D USCT system for breast cancer imaging. Figure 1 shows the prototype device with a magnified view of the measurement container. The container is filled with water to ensure acoustic impedance matching of the ultrasound transducers with the skin. The patient lies prone on top of the device and the breasts are measured individually in sequential order. Similar to magnetic resonance imaging, the data acquisition and the subsequent image reconstruction are decoupled [6].

The semi-ellipsoid measurement container holds 2041 piezoelectric transducers clustered in 157 TAS. This results in a sparse 3D imaging aperture which surrounds the immersed breast. With this arrangement, multiple imaging modalities such as transmission and reflection can be obtained simultaneously [7]. Only one transducer emits at a time, while the others record. In contrast to phased array approaches, the transducers emit unfocused sound fields. Focused reflectivity images

are obtained with the synthetic aperture focusing technique. Speed of sound and absorption images can be reconstructed using state-of-the-art algorithms modified to our environment [8].

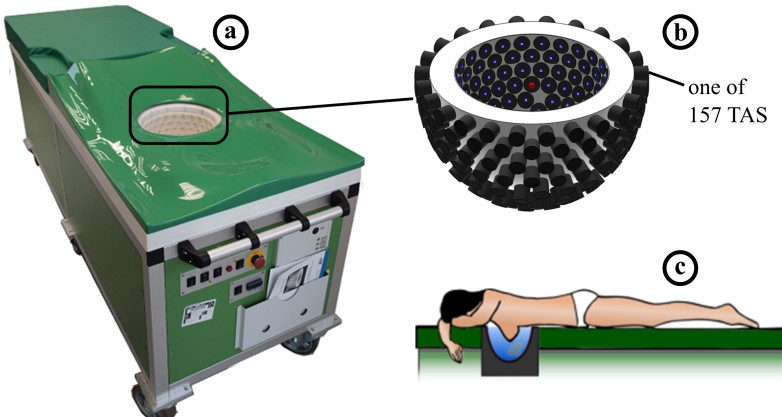

**Figure 1.** (**a**) Second generation 3D USCT system; (**b**) schematic drawing of the semi-ellipsoid measurement container which holds 157 TAS in a cylindrical housings; and (**c**) the patient position during the scan with one breast immersed in the water filled measurement container.

A clinical pilot study with 10 patients was conducted, which proved the feasibility of the 3D approach. The results of this study triggered optimisations regarding image quality and data acquisition [9]. On transducer level, this results in increasing the ultrasound emission bandwidth and opening angle. In addition, a more random transducer distribution in the aperture is desirable to reduce imaging artefacts. These considerations led to a new transducer array design based on single-fibre piezocomposites [10].

Using single-fibre piezocomposite instead of monolithic ceramics offers several advantages. Besides higher electro-mechanical coupling and lower acoustic impedance [11,12], the fibres can be arbitrary placed in the composite [13]. In addition, the round shape offers superior acoustic emission characteristics for our unfocused imaging approach [10].

To integrate the new transducers, a new assembly process is needed. This work presents a transducer array manufacturing process which encompasses adhesive layer printing, automated pick and place as well as etching techniques. For quality control, the EMI was measured at several instances which allows the assessment of not only the transducers but also the surrounding materials [14]. Several properties can then be extracted from the measured curves to allow statistical analyses.

To evaluate and predict the measured responses, simple 1D network models were used. Due to their low complexity, they allow rapid performance prediction and assist in evaluating the transducer's behaviour [15]. At first, we set up a well established model from literature [16] to match our initial transducer response. Then, we extended this basic model by adding additional layers to simulate the effects on the EMI introduced by each assembly step. This approach allows the analysis of each assembly step individually.

To ensure uniform quality of the manufactured transducers, a suitable pass/fail criterion is finally needed. For that, the extracted EMI properties can be analysed for their possibility to identify manufacturing defects. The most specific property can then be chosen to perform quality control measures during the production.

## 2. Materials and Methods

In this section, at first an overview of the new TAS is presented where each functional layer and its interconnections are briefly described. Afterwards the transducer array is explained in more detail, followed by the manufacturing process. Then, the used analytical model is introduced. Finally, the procedure of fitting the simulated initial transducer response to measured data is described.

### 2.1. Overview

The previous TAS generation was based on a dice and fill process of lead-zirconium-titanate (PZT) slabs. Each consisted of 13 quadratic transducer elements with 0.9 mm side length. One array contained four emitters and nine receivers, which were distributed in a square grid at the center of the transducer. The electrical connection was done with wire bonds onto a substrate [17]. After one year in operation, up to one third of the TAS showed element failures. A investigation revealed problems with loose wire bonds. Those were likely caused by high shear stress of the PZT [18].

For the new TAS generation, the group decided on integrating 17 transducers simultaneously in one array. This poses an acceptable trade off between hardware complexity and sparsity of the aperture. Figure 2a shows the functional TAS layers. A printed circuit board (PCB) is used for obtaining the electrical connection to the transducer layer. On top, the acoustic matching is done using thermoset microwave material (TMM4, Rogers Corp, Chandler, AZ, USA). To obtain high transducer damping, a polyurethane-tungsten compound is dispensed into boreholes foreseen in the PCB. Figure 2b shows a stacked TAS.

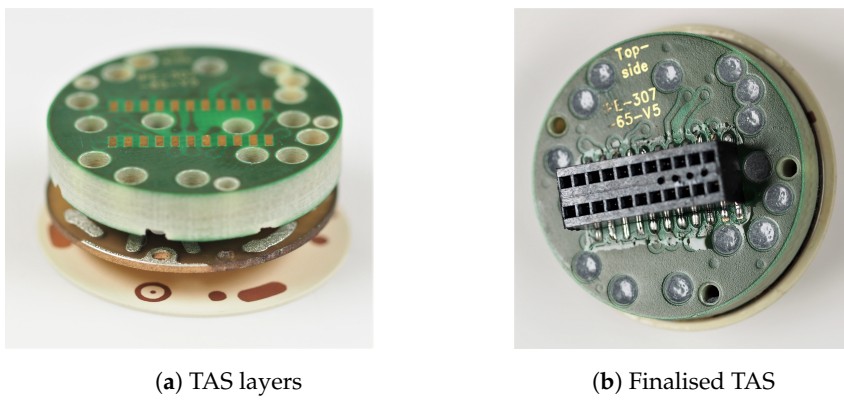

(**a**) TAS layers      (**b**) Finalised TAS

**Figure 2.** One TAS before and after the assembly: (**a**) PCB, PZT fibre disk and matching layer in stacked view (top to bottom); and (**b**) finalised TAS before attaching the housing.

The work-flow for the transducer assembly, modelling and analysis is shown in Figure 3. The EMI was measured at four instances during the assembly. This allows the evaluation of each assembly step individually. The initial measurement was used to set up a basic model, which covers the piezoelectric fibre and the bottom electrode. This basic model was then extended for each assembly step.

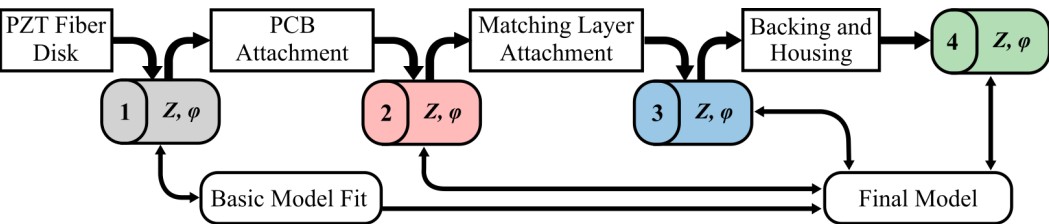

**Figure 3.** Work-flow for transducer assembly, modelling and analysis. The EMI magnitude and phase was measured at four instances. A basic model was set by fitting the model response to the initial PZT fibre response. The model is then extended by adding additional layers which simulate the following assembly steps.

### 2.2. PZT Fibre Disc

The centrepiece of the new TAS are single-fibre piezocomposites arranged in a polymer disc of 24.6 mm diameter and 0.75 mm thickness (see Figure 4). Each disc contains 17 fibres with a fibre diameter of 450 μm. The chosen dimensions were previously evaluated to optimise the emission characteristics [10]. The fibres are pseudo-randomly distributed on the disc. Simulation

predicted this distribution to be beneficial in reducing aliasing effects in the reconstructed image [19]. In addition, each disc features three holes which act as mounting aids (pinholes), two holes for ground connection (sickle-shaped electrodes at the rim) and one for placing a temperature sensor (bigger, more centrally aligned).

The fibres are produced with a polysulfone spinning process from commercial PZT powder (Sonox P505, CeramTec, Plochingen, Germany) [20]. Single fibre strands are then positioned into a mask according to the required pattern. This arrangement is placed into a mould and filled with epoxy adhesive (EpoTek 301-2, Epoxy Technology Inc., Billerica, MA, USA). After curing, the fill is machined into shape and diced. More details of the PZT fibre disc manufacturing are given in [13].

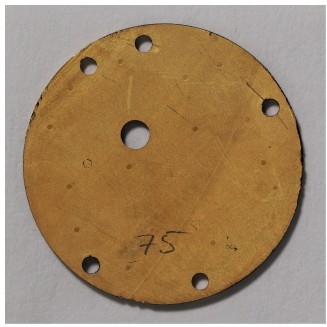 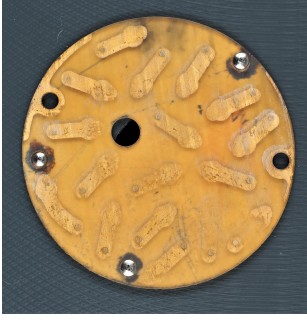 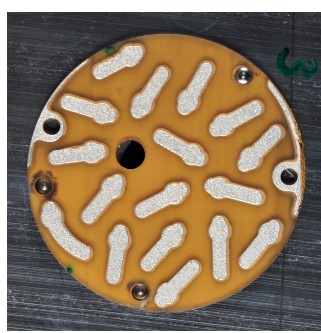

(**a**) PZT fibre disc top electrode   (**b**) Sputterd bottom electrode   (**c**) Reprinted electrodes

**Figure 4.** Preparation of the PZT fibre disc: (**a**) top view of the uniform sputtered ground electrode; (**b**) bottom view of the PZT fibre disc with sputtered electrodes which peeled off during cleaning; and (**c**) electrode print using conductive adhesive.

To electrically connect the transducers, one side of the disc is uniformly gold sputtered (see Figure 4a). The other side contains anchor-shaped bottom electrodes to allow separated conductive connection and acoustic backing (see Figure 4b). After surface preparation (cleaning with Isopropanol and wipe), the bottom electrodes showed abrasions, which led to conductive failures. Therefore, a 50-µm conductive adhesive electrode layer (EJ2189-LV, Epoxy Technology Inc., Billerica, MA, USA) was printed to the bottom of the PZT fibre disc (see Figure 4c). To avoid twisting caused by thermal mismatch, the adhesive was cured at room temperature.

### 2.3. Transducer Assembly

At first, the PZT fibre disc is connected with the PCB. The PCB holds 19 round-shaped pads for connecting the transducers, and one 0402 socket for attaching a miniaturised temperature sensor. To avoid short circuits caused by displaced adhesive, a concentric groove is milled around each pad. Seventeen drill holes are located behind the transducers after the assembly (see Figure 5a). With a depth of 4.5 mm, these drill holes hold space for the backing material. Three more drill holes are used for pinhole mounting [21]. Conductive adhesive is stencil printed to the pads (see Figure 5b) and the PZT disc then automatically placed on the PCBs (Datacon EVO2200, Besi, Radfeld, Austria). The next assembly step contains manual soldering of the temperature sensor into the designed recess. Then, the connection of the ground electrode by filling the designated holes with conductive adhesive (see Figure 5c) is done. As a safety margin, two redundant ground connections are foreseen.

One single layer aids in matching the acoustic impedance of the piezoceramic material to water. The TMM4 material and its thickness was previously evaluated to obtain suitable acoustic matching and bandwidth [17]. TMM4 is available in sheets with a designed thickness of 425 µm and a 35-µm copper layer on top. After milling the disc shape, the copper layer is structured ($FeCl_3$ wet etching, see Figure 6a). This ensures a uniform and parallel adhesive bond between the TMM4 and the PZT fibre disc.

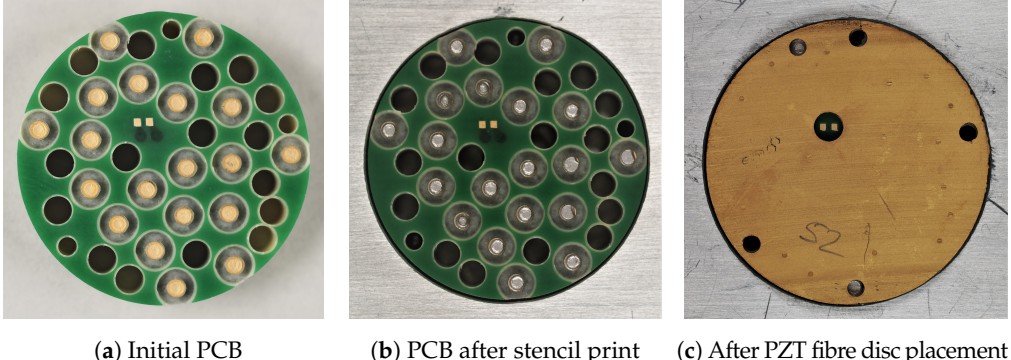

(**a**) Initial PCB      (**b**) PCB after stencil print      (**c**) After PZT fibre disc placement

**Figure 5.** (**a**) Conductive conduction process of PCB - PZT fibre disc; (**b**) stencil printing conductive adhesive onto the designated pads; and (**c**) automatic pick and place of the PZT fibre disc.

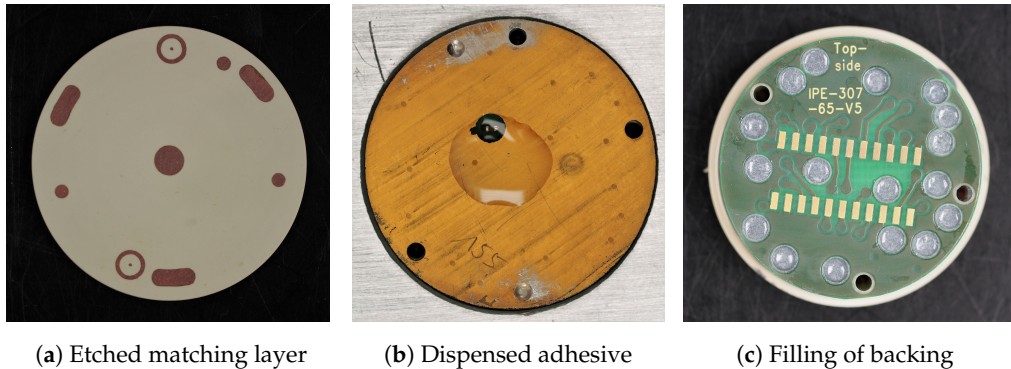

(**a**) Etched matching layer      (**b**) Dispensed adhesive      (**c**) Filling of backing

**Figure 6.** (**a**) Finalising the TAS assembly by etching the copper layer of the TMM4 disc; (**b**) dispensing 42 mg of adhesive on top of the PZT fibre disc for pick and place of the matching layer; and (**c**) application of the backing material.

For bonding the matching layer, 42 mg of adhesive (EPO-Tek 301-2) is dispensed to the centre of the PZT fibre disc (see Figure 6b). For acoustic backing, polyurethane (Flexovoss K6S, Vosschemie, Uetersen, Germany) is mixed with tungsten powder with a weight ratio of 1:2.5. This compound is then dispensed into the drill holes of the PCB (see Figure 6c). Finally, a plug is soldered to the PCB and the TAS are glued (Bondit B-45, Reltec Llc, Santa Rosa, CA, USA) into cylindrical stainless steel housings. Figure 7 shows the finalised TAS.

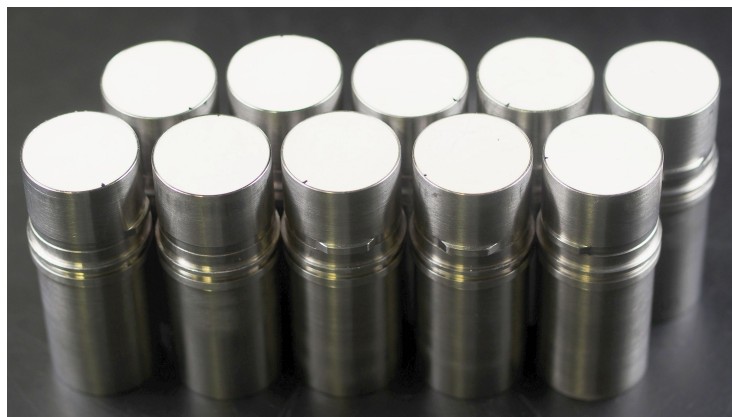

**Figure 7.** Finalised set of 10 TAS in stainless steel housing (30 mm diameter and 70 mm length). On top, the front side of the matching layer (white surface) is visible.

### 2.4. Transducer Model

To simulate the transducer response, we used the KLM equivalent circuit approach [16]. It allows rapid performance prediction based on lumped parameters [15,22]. Each assembly step can be addressed by changing or adding elements to the circuit. The model covers only linear, 1D oscillations. Hence, the obtainable results lack effects coming from other dimensions, anisotropy and non-linearities.

The PZT fibre is modelled as an ideal transformer coupled to the centre of a transmission line, as shown in Figure 8. $C_0$ is the blocking capacitance, while $C'$ and $\Phi$ are the reactance and the transforming ratio of the PZT fibre, both dependent on the excitation frequency. To account for dielectric and elastic losses of the PZT fibre, the loss tangent $\tan(\delta)$ and material attenuation $\alpha_0$ were included according to Castillo et al. [23]. $C_{\mathrm{par}}$ accounts for parasitic capacitances of the supply lines and the PCB. The transducer radiates to the front and back into a medium with acoustic impedance $Z_{\mathrm{F}}$ and $Z_{\mathrm{B}}$.

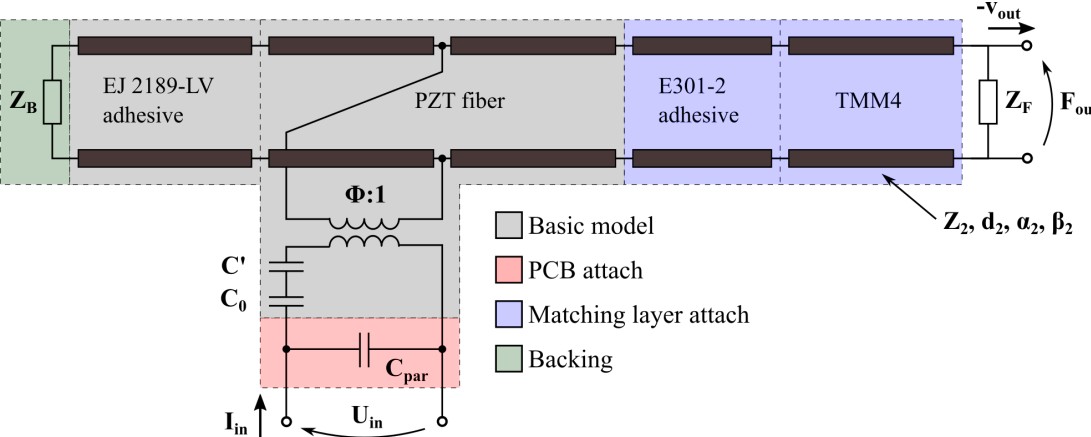

**Figure 8.** Equivalent circuit model of our transducer, based on the KLM model. The fibre and the conductive electrode form the basic model. This is then extended by adding transmission lines for each new layer.

Each of the added assembly layers are modelled with a separate transmission line. $Z$ is thereby the acoustic impedance, $d$ the thickness, $\beta$ the wave number and $\alpha$ the attenuation of the respective layer. To simplify calculations, we divided the model into transmission matrices as proposed by Kervel and Thijssen [24]. The additional layers were then added by multiplying two-ports to the PZT fibre model. All formula and more detailed descriptions can be found in Appendix A.

The initial parameters of the PZT fibre were taken from the datasheet. The acoustic parameters $Z$, $\alpha$ and $v$ of the added layers were taken, if possible, from literature [25,26]. Missing parameters were estimated. The thickness dimensions were measured with a micrometer gauge and the average layer thicknesses of $n = 15$ samples were used for the model. All parameters are listed in Appendix B.

### 2.5. Initial Model Fit

To compare measured and modelled transducer responses, the EMIs of unprocessed PZT fibre discs were measured with an impedance analyser (HP4191A, Hewlett Packard). For quantitative comparison, four properties were obtained from the measurements.

- The series resonance $f_{\mathrm{s}}$ of the thickness resonator resonance.
- The EMI magnitude $Z_{\mathrm{min}}$ at $f_{\mathrm{s}}$.
- The approximated parallel capacitance $C_{\mathrm{P}} = 1/j\omega X_{\mathrm{C}}$, calculated at 500 kHz. With phase angles close to $-90°$, the transducers behave almost purely capacitive at this frequency.
- The maximum phase angle $\varphi_{\mathrm{t}}$ of the thickness resonator resonance.

As model optimisation metric, the PZT fibre parameters were adjusted until $f_s$ and $C_p$ were within 5% of the measured average (see Table 1 for the quantitative comparison at the initial step). Figure 9a shows magnitude and phase of 68 transducers (four PZT fibre discs). This sample size was chosen to be sufficient for estimating average responses. Bold curves are measured data, while the dashed blue curves are the model predictions. The grey areas indicate all measured EMI variations. There, the centre curve is not the measured average but one single transducer, chosen for its good accordance with the initial average data.

**Table 1.** EMI properties (average and standard deviation) of four PZT fibre discs (68 transducers) initially and after each assembly step. In addition, data from the model responses are listed.

| Assembly | $f_s$ (MHz) | | $C_p$ (pF) | | $Z_{min}$ (kΩ) | | $\varphi_t$ (°) | |
|---|---|---|---|---|---|---|---|---|
| **Step** | meas. | mod. | meas. | mod. | meas. | mod. | meas. | mod. |
| (1) Initial | $1.84 \pm 0.04$ | 1.86 | $3.35 \pm 0.23$ | 3.23 | $14.18 \pm 1.45$ | 13.38 | $-3.12 \pm 9.08$ | 2.61 |
| (2) PCB | $1.83 \pm 0.04$ | 1.86 | $5.42 \pm 0.92$ | 4.96 | $11.19 \pm 1.34$ | 11.27 | $-31.56 \pm 10.26$ | $-28.68$ |
| (3) Matching L. | $1.94 \pm 0.05$ | 2.04 | $5.25 \pm 1.06$ | 4.99 | $13.42 \pm 2.06$ | 12.71 | $-45.54 \pm 14.13$ | $-45.56$ |
| (4) Final | $1.94 \pm 0.05$ | 2.04 | $6.87 \pm 0.89$ | 6.26 | $11.74 \pm 1.40$ | 12.80 | $-69.27 \pm 4.83$ | $-69.22$ |

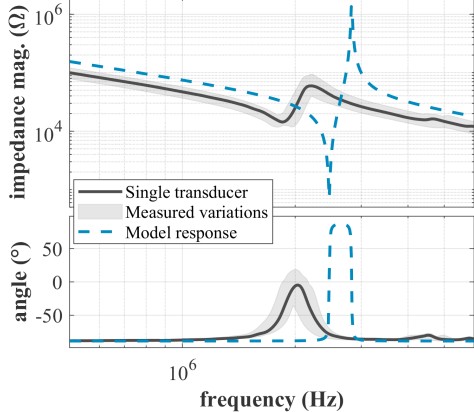 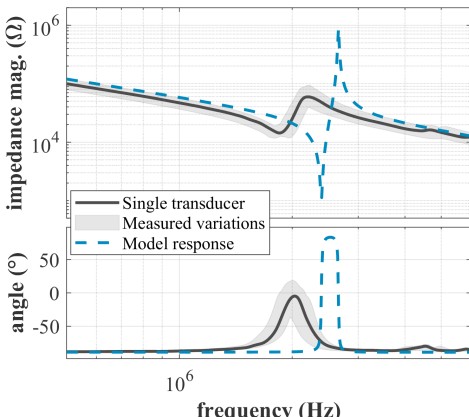

(**a**) Data comparison with initial model parameters

(**b**) After adding the backing layer (conductive adhesive) and a small supply line capacitance

**Figure 9.** Measured and modelled transducer response before (**a**) and after (**b**) adding the electrode. Sixty-eight transducers were measured. The grey area shows the total measurement variations and the bold curve one single transducer response, chosen for its good accordance with the average.

Using the parameters from the datasheet, the model predicts $f_s$ at 2.46 MHz. This differs by 0.54-MHz from the measured average. To investigate this difference, $f_s$ was calculated using the thickness frequency constant $N_t$. This results according to Equation (1) in 2.44 MHz for $f_s$. As the calculated value also differs from the measured average, a change in material properties caused by the PZT fibre fabrication process may have occurred.

$$f_s = \frac{N_t}{d_0} \tag{1}$$

To improve the prediction accuracy, at first, the conductive adhesive electrode was added to the model. Then, the supply line capacitances $C_{par}$ was set to 0.6 pF, accounting for the electrode on the PZT fibre disc. As shown in Figure 9b, adding these parameters reduced the EMI offset, but the difference in $f_s$ remained almost constant.

The series resonance $f_s$ is predominantly affected by the elastic stiffness $c_{33}^D$. When changing this parameter, it is important to consider the dependent character of piezoelectric material properties. By using the parameter set from the materials datasheet in strain-charge form, $c_{33}^D$ can be derived

according to Chevallier et al. [27] as dependent parameter. Increasing the compliance $s_{33}^E$ then causes a decrease in the elastic stiffness and rises the dielectric constant $\epsilon_{33}^S$.

The thickness coupling factor $k_t$ also affects the position of $f_s$ on the frequency axis. As previously reported by Hohlfeld et al. [20], the PZT fibre fabrication results in an increase of $k_t$ by 10–20%, compared to bulk PZT discs. Increasing $s_{33}^E$ by 11% and $k_t$ by 9% resulted in the response shown in Figure 10a. Now, the position of $f_s$ matches the measured data, but still high differences in damping are visible.

By increasing the attenuation coefficient $\alpha_0$ and the backing impedance $Z_B$ significantly, the transducer resonates according to Figure 10b. This additional damping may not represent the PZT fibre behaviour correctly, as lateral effects are inherently neglected with this model. The initial and adjusted model parameters are listed in Table A1.

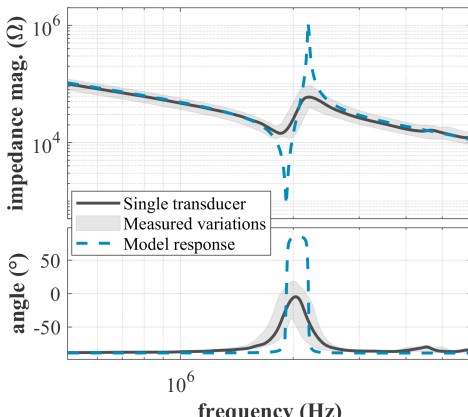 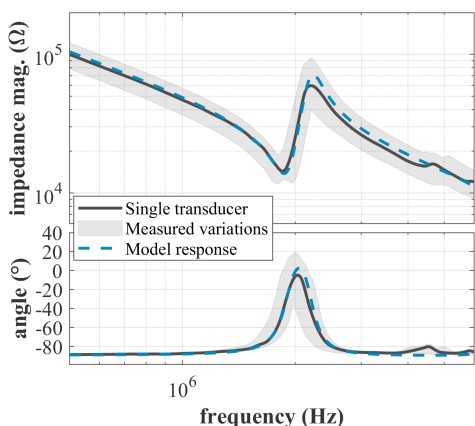

(**a**) Transducer response after adjusting the compliance and coupling　　(**b**) Final model fit after increasing the attenuation

**Figure 10.** (**a**) Measured and modelled transducer response after increasing the compliance $s_{33}^E$ by 11% and the thickness coupling $k_t$ by 9%; and (**b**) final model fit after increasing the attenuation $\alpha_0$ and backing impedance $Z_B$.

## 3. Results

### 3.1. Model Predictions

Adding the PCB should only increase the supply line capacitance. Therefore, the basic model can be extended by setting $C_{par}$ to 2.5 pF. The responses of the transducer samples ($n = 68$) and the model are shown in Figure 11a. The bold curve is the response of the same single transducer which was used for the model fit. As shown, the EMI magnitude and the angle at resonance decrease, compared to the initial state (see Figure 9a). These effects are predicted well with the model.

Adding the two matching layers is modelled by setting the respective thicknesses of the transmission lines. The resulting response is given in Figure 11b. The model prediction shows prominently an additional resonance at 1.28 MHz, introduced by the thickness of the TMM4 layer. In addition, it predicts a slight rise in stiffness resulting in a shift of $f_s$ to 2.04 MHz. The measurements support this behaviour with a shift of $f_s$ to 1.94 MHz, although the additional resonance is not present.

The absence of the predicted matching layer resonance is most likely not caused by wrong material parameters, as variations of those did not lead to convincing results (see Figure 12b). One possible explanation could be the size difference between the matching layers and the PZT fibre, as only one TMM4 disc is used for all 18 transducers.

For the last assembly steps, the model is at first adjusted by setting only $Z_B$ to 5.8 MPas/m. This accounts for the acoustic backing. The resulting response is shown in Figure 12a and indicates a consistent EMI offset compared to the measured data.

The lack of the additional resonance induced by the matching layer can be investigated with the model. Altering the speed of sound $v_2$ and characteristic impedance $Z_2$ within justifiable limits ($\pm25\%$) did not suppress the additional resonance. However, by increasing the attenuation coefficient $\alpha_2$, the effect of the matching layer on the model response can be suppressed. In Figure 12b, the model response after tripling $\alpha_2$ is shown. In addition, $C_{par}$ was increased to compensate the EMI offset. While the modelled magnitude levels better with the measured responses after the adjustment, the phase angle diverges for the transducers thickness resonance. The final transducer model parameter set is listed in Table A2.

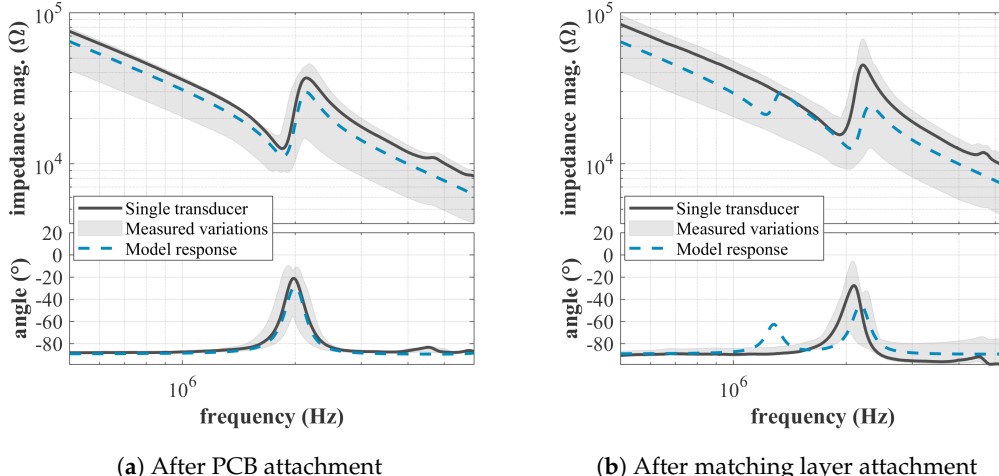

(**a**) After PCB attachment    (**b**) After matching layer attachment

**Figure 11.** Measured and modelled EMI after attaching: the PCB (**a**); and the matching layers (**b**). For modelling the PCB attachment, $C_{par}$ was increased. The matching layers are added to the model by setting the measured thickness of the respective transmission lines.

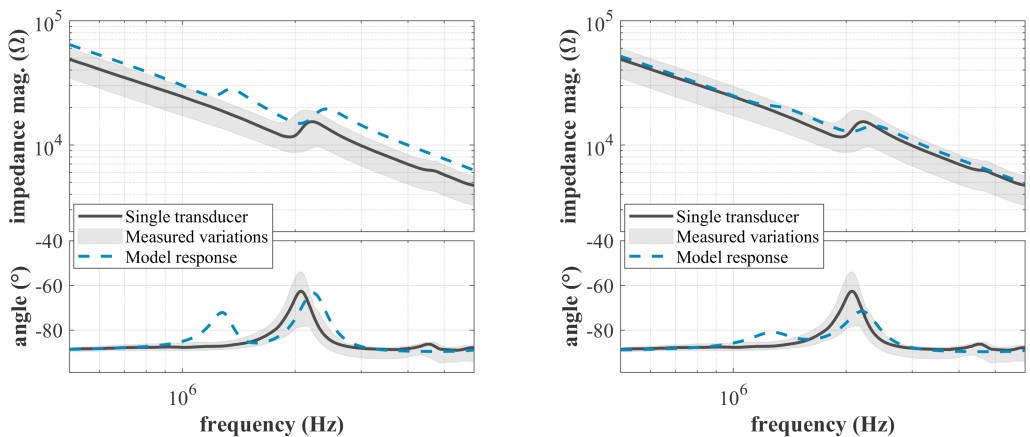

(**a**) Final response without parameter adjustment    (**b**) Supply line capacitance and matching layer attenuation increased

**Figure 12.** Final transducer EMI with at first only increased backing impedance $Z_B$ (**a**). The magnitude offset is compensated by increasing $C_{par}$. The induced additional resonance can be suppressed by increasing the attenuation coefficient $\alpha_2$ (**b**).

### 3.2. Assembly Analysis

Table 1 summarises the four EMI properties introduced in Section 2.5 at the four assembly stages. The column next to the measured average and standard deviation shows the modelled responses. The initial state represents data from blank PZT fibre discs. The next rows correspond to data after the PCB and the matching layers are attached. The final dataset was measured after applying the backing material and the housing.

Comparing the properties before and after the assembly, the series resonance $f_s$ slightly rises. $C_p$ more than doubles with a significant rise in standard deviation. $Z_{min}$ falls by 17%, whereby the highest deviation was measured after attaching the matching layer. The most significant change is the phase angle $\varphi_t$. It drops by more than 60° and shows less standard deviation after the assembly, compared to the initial state.

Stepwise, attaching the PCB to the PZT fibre discs does not affect the resonating behaviour, as $f_s$ remains constant. It only adds a parallel capacitance of in average 2.07 pF. The increase in standard deviation of $C_p$ is caused by the routing of the PCB. Adding the matching layer in the next step induces only small variations of the measured parameters, but shifts $f_s$ already to the final value. Casting the backing finally causes an additional rise in $C_p$ and a strong drop of $\varphi_t$.

### 3.3. Quality Control

Analysing the EMI properties can now be used to monitor the quality of the assembly. Two possible defects, which cannot be detected optically, are air inclusions in front and behind the PZT fibre. This would either cause low emission due to high reflections or low damping resulting in a decreased bandwidth.

The phase angle at thickness resonance $\varphi_t$ seems the most suitable property for monitoring the production and identifying these two defects. The other three EMI properties listed in Table 1 exhibit high overlapping between the measured distributions. Only $\varphi_t$ shows a consistent change after each assembly step. However, due to the high standard deviation after attaching the PCB and the matching layer, unambiguous classification may not be possible.

To visualise all possible classifications for detecting the backing and the matching layer attachment, a receiver operating characteristic curve (ROC) can be used. There, the effect of potential classifiers on false and true positive decisions are visualised [28]. Figure 13a shows the ROC curves after attaching the matching layer and the backing for $-80° \leq \varphi_t \leq 0°$. The matching layer ROC exhibits a lower trend compared to the backing ROC, which indicates poorer classifying performance.

When allowing a maximum of 5% false positive decisions, detecting a successful attachment of the matching layer results in the classifier T1 at $-50°$. Likewise, detecting if the backing is applied correctly gives T2 at $-62.6°$. With a resulting true positive detection rate of 94% for T2, it is possible to find faults in the backing reliably. This is not given for T2, as correct matching layer attachments can be identified with only 46% accuracy. All measured phase angles $\varphi_t$ and the two classifiers are visualised in Figure 13b.

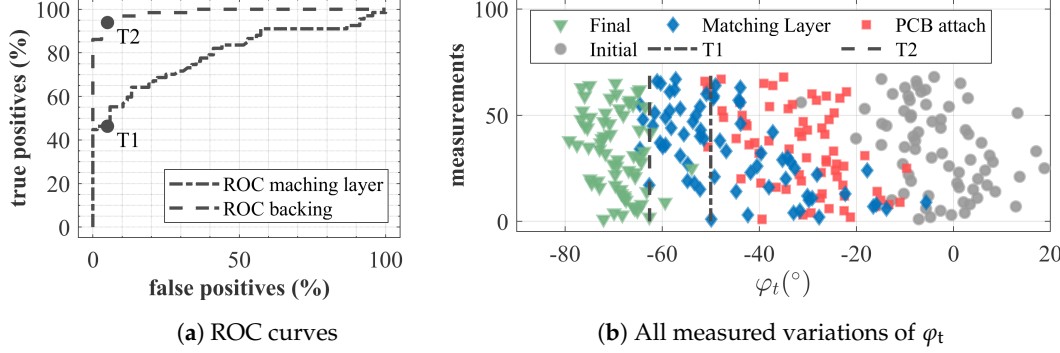

(**a**) ROC curves　　　　　　　　　　(**b**) All measured variations of $\varphi_t$

**Figure 13.** Quality control using the measured phase angle at thickness resonance $\varphi_t$ as classifier to detect correct matching layer and backing application. ROC curves for $-80° \leq \varphi_t \leq 0°$ and classifiers T1 and T2 at maximum 5% false positive decisions (**a**). T2 indicates good performance for backing detection. T1 shows less than 50% sensitivity for detecting correct matching layer attachment (**b**).

## 4. Discussion

The proposed assembly process allows reliable and time efficient manufacturing of our new TAS generation. The resulting transducers show a series resonance frequency $f_s$ at $1.94 \pm 0.05$ MHz. The low standard deviation suggests also low deviations in acoustic performance, which is important for our imaging approach.

After the assembly, the transducer capacitance $C_p$ more than doubled. This is mainly caused by an additional supply line capacitance coming from the PCB. However, the increase in $C_p$ from the matching layer to the final state cannot directly be explained by an additional electrical capacitance, as no conductor is added. Due to the electromechanical coupling of the PZT fibre, this can be caused by electrical or mechanical effects. In the electrical domain, a change in permittivity caused by the tungsten powder may be an explanation. In the mechanical domain, a change in stiffness also affects the capacitance, but mostly for frequencies close to $f_s$.

With the extended KLM model, the change in EMI caused by attaching the PCB can be predicted. In addition, the effect of the backing on the transducer damping is covered accurately. Simulated effects induced by the matching layers do not match the measured responses. Especially the predicted additional resonance caused by the thickness of the single TMM4 matching layer is not present in the measured data. More dimensional modelling approaches seem necessary to investigate this difference. However, due to the low complexity of the proposed model, parameter evaluation can be done very quickly and easily. The obtained results can then be used as a starting point for more computational expensive modelling approaches such as finite element methods.

The transducer condition during production can be monitored most effectively by measuring the phase angle $\varphi_t$ at thickness resonance. Ninety-four per cent of all backing defects should be detectable by setting the pass/fail classification criteria to $-62.6°$. For detecting defects in attaching the matching layer, a sensitivity of less than 50% can be achieved. To improve this, more EMI properties must be taken into account. Using K-means or hierarchical clustering methods [29], higher specificity should be achievable. Future work will therefore focus on finding the most sensitive property combination for ensuring a reliable quality control procedure.

**Author Contributions:** Conceptualization, M.A., M.Z. and N.V.R.; methodology, M.A. and M.Z.; validation, M.A.; resources, B.L.; writing—original draft, M.A.; and writing—review and editing, M.Z. and N.V.R. All authors have read and agreed to the published version of the manuscript.

**Funding:** This research received no external funding.

**Acknowledgments:** The authors thank Paul Günther from Fraunhofer IKTS for his valuable support in evaluating the PZT fibre disc performance. We acknowledge support by the KIT-Publication Fund of the Karlsruhe Institute of Technology.

**Conflicts of Interest:** The authors declare no conflict of interest.

## Abbreviations

The following abbreviations are used in this manuscript:

| | |
|---|---|
| EMI | Electro-mechanical impedance |
| KLM | Transducer network model developed by R. Krimholtz, D. Leedom and G. Matthaei |
| PZT fibre disc | Single-fibre piezocomposite disc array |
| PCB | Printed circuit board |
| PZT | Lead zirconium titanate |
| ROC | Receiver operating characteristic curve |
| TAS | Transducer array system |
| TMM4 | Thermoset microwave material, Rogers Corp. |
| USCT | Ultrasound computer tomography |

## Appendix A. KLM Model

The implemented model can be reduced to a $2 \times 2$ transfer matrix, with electrical power ($P_{in} = U_{in} \cdot I_{in}$) at the input and mechanical (radiation) power ($P_{out} = F_{out} \cdot v_{out}$) at the output. It connects the input and output magnitudes using voltage-to-force conversion. By using the two-port formalism, the model can be divided into arbitrary sub-matrices, which are subsequently concatenated. Figure A1 shows a scheme of the used sub-matrices, where each block correlates to a transferred impedance. As indicated, the backward radiation is connected in parallel with the forward radiation.

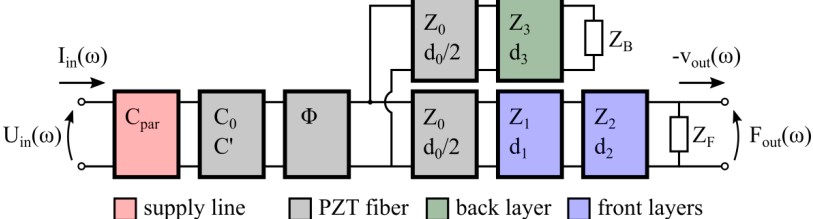

**Figure A1.** Scheme of the transducer model with eight concatenated two-ports. Each port represents a transferred impedance. The backward radiation is coupled in parallel to the forward radiation.

The blocking capacitance $C_0$ is calculated according to Equation (A1), where $\epsilon_{33}^S$ is the permittivity at constant strain, $A_0 = r_0^2 \pi$ the electrode surface area and $d_0$ the thickness of the fibre. Dielectric losses are considered by adding the imaginary part $j\tan(\delta)$ to the permittivity, which includes the loss factor $\delta$ of the material.

$$C_0 = \epsilon_{33}^S (1 - j\tan(\delta)) \cdot \frac{A_0}{d_0} \tag{A1}$$

The acoustic impedance $Z_0$ of the fibre material is determined using Equation (A2), where $\rho$ is the density and $v_0$ the materials speed of sound. Due to the anisotropic behaviour of the material, the speed of sound can only be approximated in thickness direction according to Equation (A3). There, $c_{33}^D$ is the elastic stiffness at constant dielectric displacement.

$$Z_0 = \rho \cdot v_0 \tag{A2}$$

$$v_0 = \sqrt{\frac{c_{33}^D}{\rho}} \tag{A3}$$

The fibre material is modelled with a transformer coupled to the centre of a transmission line (see Figure 8). It converts electrical input power with a frequency dependent turns ratio $\Phi(\omega)$ to a mechanical wave. A frequency dependent capacitance $C'(\omega)$ acts as reactance, transformed to the primary side of the transformer. The used analytical expressions for the turns ratio and the reactance are given in Equation (A4) and Equation (A5). More detailed descriptions of this modelling approach can be found in [30].

$$\Phi(\omega) = k_t \cdot \sqrt{\frac{d_0}{v_0 C_0 Z_0}} \cdot \text{sinc}(\omega/2\omega_0) \tag{A4}$$

$$C'(\omega) = \frac{-C_0}{k_t^2 \cdot \text{sinc}(\omega/\omega_0)} \quad \text{with} \quad \omega_0 = \pi v_0 / d_0 \tag{A5}$$

To account for backward radiation, the impedance of the back-side medium $Z_B$ must be transferred to the centre of the PZT fibre. This is done in two steps. First, $Z_B$ is transferred to the input of the backing layer $Z_3$ according to Equation (A6). Second, the resulting impedance $Z_{B1}$ is transferred to

the centre of the PZT fibre using Equation (A7). Elastic losses are modelled by adding an attenuation coefficient $\alpha$ to the wave number $\beta$, both individually calculated for each transmission line.

$$Z_{B1}(\omega) = Z_3 A_0 \cdot \frac{Z_B + j(Z_3 \cdot \tan(\beta_3 d_3))}{Z_3 + j(Z_B \cdot \tan(\beta_3 d_3))} \quad \text{with} \quad \beta_3 = \frac{\omega}{v_3}(1 - j\alpha_3 d_3) \tag{A6}$$

$$Z_{PB}(\omega) = Z_0 A_0 \cdot \frac{Z_{B1} + j(Z_0 \cdot \tan(\beta_0 d_0/2))}{Z_0 + j(Z_{B1} \cdot \tan(\beta_0 d_0/2))} \quad \text{with} \quad \beta_0 = \frac{\omega}{v_0}(1 - j\alpha_0 d_0/2) \tag{A7}$$

The overall transmission matrix of the transducer is then calculated according to Equation (A8), where each $2 \times 2$ matrix corresponds to one of the two-ports shown in Figure A1. The wave numbers $\beta_1$ and $\beta_2$ are calculated according to Equation (A6), only with their respective thickness, velocity and attenuation coefficient.

$$
\begin{aligned}
T(\omega) = \begin{bmatrix} A(\omega) & B(\omega) \\ C(\omega) & D(\omega) \end{bmatrix} &= \begin{bmatrix} 1 & 0 \\ j\omega C_{par} & 1 \end{bmatrix} \cdot \begin{bmatrix} 1 & 1/j\omega C_0 + 1/j\omega C'(\omega) \\ 0 & 1 \end{bmatrix} \\
\cdot \begin{bmatrix} \Phi(\omega) & 0 \\ 0 & 1/\Phi(\omega) \end{bmatrix} \cdot \begin{bmatrix} 1 & 0 \\ 1/Z_{PB}(\omega) & 1 \end{bmatrix} &\cdot \begin{bmatrix} \cos(\beta_0 d_0/2) & jZ_0 A_0\sin(\beta_0 d_0/2) \\ j\sin(\beta_0 d_0/2)/Z_0 A_0 & \cos(\beta_0 d_0/2) \end{bmatrix} \\
\cdot \begin{bmatrix} \cos(\beta_1 d_1) & jZ_1 A_0\sin(\beta_1 d_1) \\ j\sin(\beta_1 d_1)/Z_1 A_0 & \cos(\beta_1 d_1) \end{bmatrix} &\cdot \begin{bmatrix} \cos(\beta_2 d_2) & jZ_2 A_0\sin(\beta_2 d_2) \\ j\sin(\beta_2 d_2)/Z_2 A_0 & \cos(\beta_2 d_2) \end{bmatrix}
\end{aligned} \tag{A8}
$$

The $2 \times 2$ matrix allows now the derivation of transducer specific characteristics. One important parameter is the EMI, seen by the driving power source. It can be obtained by enforcing the boundary condition $F_{out} = Z_F A_0 \cdot v_{out}$, which collapses the two-port network to the expression given in Equation (A9).

$$Z_{in}(\omega) = \frac{U_{in}}{I_{in}(\omega)} = \frac{A(\omega) \cdot Z_F A_0 + B(\omega)}{C(\omega) \cdot Z_F A_0 + D(\omega)} \tag{A9}$$

## Appendix B. Model Parameters

**Table A1.** Parameters of PZT fibre material (CeramTec SONOX P505) and dimensions for the basic model. Initial values are taken from the datasheet or otherwise obtained. The optimised values are fitted to measured responses.

| Parameter | Unit | Initial Step | Optimised |
|---|---|---|---|
| $d_0$ | μm | 770 | 770 |
| $r_0$ | μm | 450 | 450 |
| $k_t$ | – | 0.53 | 0.58 |
| $s_{33}^E$ | m$^2$/N | $24.0 \times 10^{-12}$ | $27.1 \times 10^{-12}$ |
| $\epsilon_{33}^T/\epsilon_0$ | – | 1880 | 1880 |
| $\alpha_0$ | 1/m | 6.28 | 130 |
| $tan(\delta)$ | – | 0.02 | 0.02 |
| $Z_B$ | MPa·s/m | $0.4 \times 10^{-3}$ | 0.8 |
| $Z_F$ | MPa·s/m | $0.4 \times 10^{-3}$ | $0.4 \times 10^{-3}$ |
| $C_{par}$ | pF | 0.0 | 0.50 |

**Table A2.** Parameters of the extended transducer model in the final step and after optimisation. The acoustic parameters with subscript 1 relate to the front adhesive layer (EpoTek 301-2), subscript 2 to the second front layer (TMM4 material) and subscript 3 to the conductive epoxy electrode (EpoTek EJ2189-LV).

| Parameter | Unit | Final Step | Optimised |
|:---:|:---:|:---:|:---:|
| $d_1$ | μm | 60 | 60 |
| $\alpha_1$ | 1/m | 298 | 298 |
| $Z_1$ | MPa·s/m | 2.85 | 2.85 |
| $v_1$ | m/s | 2640 | 2640 |
| $d_2$ | μm | 430 | 430 |
| $\alpha_2$ | 1/m | 157 | 471 * |
| $Z_2$ | MPa·s/m | 6.40 | 6.40 |
| $v_2$ | m/s | 3280 | 3280 |
| $d_3$ | μm | 50 | 50 |
| $\alpha_3$ | 1/m | 631 | 631 |
| $Z_3$ | MPa·s/m | 5.14 | 5.14 |
| $v_3$ | m/s | 1700 | 1700 |
| $Z_B$ | MPa·s/m | 0.3 | 5.8 |
| $Z_F$ | MPa·s/m | $0.4 \times 10^{-3}$ | $0.4 \times 10^{-3}$ |
| $C_{par}$ | pF | 0.65 | 3.70 |

* initial value was tripled to suppress effects of matching layer.

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
