# Peer review of "Model-Guided Manufacturing of Transducer Arrays Based on Single-Fibre Piezocomposites"

_applsci, doi:10.3390/app10144927_

Round 1
Reviewer 1 Report
1- The authors should reinforce the introduction of this article with new references in this area such as:
- Improvement of the vibratory diagnostic method by evolution of the piezoelectric sensor performances
- Modeling and enhancement of piezoelectric accelerometer relative sensitivity
2- Illustrate the purpose of this work in introduction
3- Correct spelling mistakes
Author Response
Dear Reviewer,
thank you very much for your remarks. We tried to implement your suggestions as follows:
1- We added the following paragraph to the introduction (line 52-56):
"Using single-fibre piezocomposite instead of monolithic ceramics offers several advantages. Besides higher electro-mechanical coupling and lower acoustic impedance [11,12], the fibres can be arbitrary placed in the composite [13]. In addition, the round shape offers superior acoustic emission characteristics for our unfocused imaging approach [10].
To integrate the new transducers, a new assembly process is needed. This work presents a transducer array manufacturing process which encompasses adhesive layer printing, automated pick and place as well as etching techniques."
2- We tried to sharpen the workflow in the abstract and introduction to:
Background ->Transducer requirements -> Transducer design -> Controlled manufacturing approach
3- We revised the article to correct wrong spelling, singular/plural errors as well as capitalisation.
Best regards

Reviewer 2 Report
This seems like an interesting and useful modelling-led technique to improve process monitoring during piezo array fabrication.
Author Response
Dear Reviewer,
thank you for the positive reply. Attached you find a slightly updated version of the paper. In the introduction, we added a paragraph to introduce the used piezocomposites in more detail (line 52-56). In addition, we made several small corrections mainly in the abstract and the introduction.
With best regards
